# How maternity waiting home use influences attendance of antenatal and postnatal care

**Julie M. Buser**[1]*, **Michelle L. Munro-Kramer**[2], **Philip T. Veliz**[3], **Xingyu Zhang**[3], **Nancy Lockhart**[2], **Godfrey Biemba**[4,5,6], **Thandiwe Ngoma**[6,7], **Nancy Scott**[5‡], **Jody R. Lori**[2‡]

**1** Michigan Medicine, University of Michigan, Ann Arbor, Michigan, United States of America, **2** Department of Health Behavior and Biological Sciences, University of Michigan School of Nursing, Ann Arbor, Michigan, United States of America, **3** Department of Systems, Populations and Leadership, Applied Biostatistics Laboratory, University of Michigan School of Nursing, Ann Arbor, Michigan, United States of America, **4** National Health Research Authority, Lusaka, Zambia, **5** Department of Global Health, Boston University School of Public Health, Boston, Massachusetts, United States of America, **6** Zambian Center for Applied Health Research and Development (ZCAHRD) Limited, Lusaka, Zambia, **7** Right to Care Zambia, Lusaka, Zambia

‡ These authors are joint senior authors on this work.
* jbuser@umich.edu

**Data Availability Statement:** All relevant data are within the paper.

**Funding:** The study program was developed and is being implemented in collaboration with Merck for

## Abstract

As highlighted in the International Year of the Nurse and the Midwife, access to quality nursing and midwifery care is essential to promote maternal-newborn health and improve survival. One intervention aimed at improving maternal-newborn health and reducing underutilization of pregnancy services is the construction of maternity waiting homes (MWHs). The purpose of this study was to assess whether there was a significant change in antenatal care (ANC) and postnatal care (PNC) attendance, family planning use, and vaccination rates before and after implementation of the Core MWH Model in rural Zambia. A quasi-experimental controlled before-and-after design was used to evaluate the impact of the Core MWH Model by assessing associations between ANC and PNC attendance, family planning use, and vaccination rates for mothers who gave birth to a child in the past 13 months. Twenty health care facilities received the Core MWH Model and 20 were identified as comparison facilities. Before-and-after community surveys were carried out. Multivariable logistic regression were used to assess the association between Core MWH Model use and ANC and PNC attendance. The total sample includes 4711 mothers. Mothers who used the Core MWH Model had better ANC and PNC attendance, family planning use, and vaccination rates than mothers who did not use a MWH. All mothers appeared to fare better across these outcomes at endline. We found an association between Core MWH Model use and better ANC and PNC attendance, family planning use, and newborn vaccination outcomes. Maternity waiting homes may serve as a catalyst to improve use of facility services for vulnerable mothers.

Mothers, Merck's 10-year, $500 million initiative to help create a world where no woman dies giving life. Merck for Mothers is known as MSD for Mothers outside the United States and Canada (MRK 1846-06500.COL). The development of the study article was additionally supported in part by the Bill & Melinda Gates Foundation (OPP1130334) https://www.gatesfoundation.org/How-We-Work/Quick-Links/Grants-Database/Grants/2015/06/OPP1130334 and The ELMA Foundation (ELMA-15-F0010) http://www.elmaphilanthropies.org/the-elma-foundation/. The funders had no role in study design, data collection and analysis, decision to publish, or preparation of the manuscript. The content is solely the responsibility of the authors and does not necessarily reflect positions or policies of Merck, the Bill & Melinda Gates Foundation, or The ELMA Foundation.

**Competing interests:** The authors have declared that no competing interests exist.

## Introduction

Access to quality nursing and midwifery care is essential to promote maternal-newborn health and improve survival. However, underutilization of maternal healthcare services in limited-resource settings is partly responsible for maternal deaths during pregnancy, childbirth or within a few weeks of giving birth [1]. Despite global progress in reducing maternal mortality, immediate action is needed to eliminate preventable maternal mortality [2].

The International Year of the Nurse and the Midwife highlighted the importance of nursing and midwifery care for maternal-newborn health and researchers can advocate for improved access to quality nursing and midwifery care [3]. Nurses and midwives are often the first and only point of care in their communities and they play a vital role in eliminating preventable maternal mortality and providing health services [3]. Thus, improving access to skilled nurses and midwives at the facility-level has the potential to improve maternal and newborn health outcomes [4]. In Zambia, the health system is organized into different levels of health facilities for service provision with maternal health services provided at the: (1) health post level (the lowest level), (2) health center level, (3) the level one (district) hospital, and (4) level two and level three (tertiary) hospitals [5].

Although maternal-newborn mortality is largely preventable, newborn and maternal health in Zambia remain poor [6]. The remote and poorest populations are the most vulnerable and marginalized communities [7]. In rural Zambia, mothers often face suboptimal care and underutilization of maternal health services [8]. Nonetheless, slow progress is being made in Zambia to improve maternal-newborn survival. From 2015 to 2017, the maternal mortality rate in Zambia decreased from 232 to 213 per 100,000 live births and infant mortality decreased from 45 to 42 per 1,000 live births [9, 10].

Utilization of antenatal care (ANC) presents a unique and lifesaving opportunity for health promotion, disease prevention, early diagnosis and treatment of illnesses in pregnancy using evidence-based practices [11, 12]. According to the 2018 Zambia Demographic Health Survey (DHS), 64% of mothers attended four or more ANC visits, a noticeable increase from the 56% in the 2013-2014 DHS [13]. Rural mothers were slightly more likely than urban mothers to have attended four or more ANC visits (65% and 61%, respectively) [13]. At the time of the study, ANC was considered inadequate if a woman has three or fewer ANC visits or does not initiate ANC in the first trimester [14]. Additionally, new ANC recommendations from WHO advocate a minimum of eight contacts to reduce perinatal mortality and improve women's experience of care [15].

Most maternal and infant deaths occur in the first month after birth, therefore, the WHO recommends all mothers and babies need at least four postnatal checkups in the first 6 weeks [16]. In 2018, 70% of Zambian women received postnatal care (PNC) within 2 days after giving birth, an increase from 63% in 2013-2014 [13, 17]. There was a large gap between women in urban (82%) and rural (64%) areas of Zambia in receiving timely PNC [13]. A lack of PNC affects the coverage of several essential interventions and missed opportunities to promote healthy behaviors, affecting women, newborns, and children [12]. Provision of maternal health education about family planning and newborn vaccinations are important components of PNC.

One intervention aimed at reducing underutilization of obstetric services and improving maternal-newborn health in Zambia is the construction or refurbishment of maternity waiting homes (MWHs) as part of the larger health system strengthening initiative, Saving Mothers, Giving Life (SMGL). Saving Mothers, Giving Life was a public-private partnership to dramatically reduce maternal and newborn mortality in Zambia and other sub-Saharan African countries [18]. In Zambia, from 2013-2018, SMGL put in place key interventions, to improve

maternal and newborn health across 16 districts in which the initiative set out to make high-quality, safe childbirth services with a skilled midwife available and accessible to women and their newborns, focusing on the critical period of labor, birth and the first 48 hours postpartum [19].

Maternity waiting homes, also known as mother's shelters, are residential facilities, located near a health facility or hospital, where women can await giving birth and be transferred to a nearby health facility staffed by a skilled midwife shortly before giving birth, or earlier should complications arise [20]. Maternity waiting homes have been used since the 1950s with positive maternal health outcomes [21]. However, a Cochrane review to assess the effects of MWHs on maternal and perinatal health recommended that well controlled trials are needed to continue to build evidence on MWH outcomes [22].

In 2015, the Maternity Homes Alliance for Zambia was formed between the Government of Zambia, donors, implementing partners, university evaluators, and the SMGL project, to provide robust data for decision-makers on the effectiveness of MWHs in Zambia as an intervention to increase access to health facilities for all pregnant women with skilled midwives and improve maternal and newborn health outcomes [23]. The pillars of the Core MWH Model implemented by the Maternity Waiting Home Alliance included: (1) infrastructure, equipment, and supplies to address the need for higher quality, safer MWHs where mothers can wait comfortably before receiving clinical care (e.g., ANC, giving birth, or PNC), (2) policies, management and financial structures, and (3) linkages to health systems with skilled midwives to ensure mothers receive appropriate ANC or PNC while waiting [24]. In terms of facility, the structures that were part of the Core MWH Model have sustainable features (e.g., concrete walls and floors, roofs that do not leak, latrines, a private bathing space, water within a reasonable distance, a covered cooking space and storage space) [24]. The Core MWH Model placed the structures close to health facilities to ensure timely access to clinical care (ANC, giving birth, or PNC) [24]. A health facility staff provided daily check-ins with waiting women but clinical care visits continued to be conducted at the health facility, not in the MWH [24]. Women staying at the Core MWH Model had the opportunity to participate in maternal and child education courses offered by the health facility staff or community health workers [24]. Finally, the Core MWH Model is community owned and operated with community health workers (Safe Motherhood Action Group members) promoting the use of the MWHs and clinical care [24].

The purpose of this analysis was to assess whether there was a significant change in ANC and PNC attendance, family planning use, and vaccination rates before and after implementation of the Core MWH Model in rural Zambia.

## Materials and methods

### Design and setting

A quasi-experimental controlled before-and-after design was used to assess associations between ANC and PNC attendance, family planning use, and vaccination rates for mothers who gave birth to a child in the past 13 months. Mothers birthed in 40 healthcare facilities in three provinces (Eastern, Luapula, and Southern) and seven districts (Chembe, Choma, Kalomo, Lundazi, Mansa, Nyimba, and Pemba) that were part of the SMGL initiative [24–26]. Twenty health care facilities received the minimum Core MWH Model and 20 were identified as comparison facilities [24, 25]. Population-level data were collected before implementation of the Core MWH Model in 2016 and after implementation in 2018. Specific details outlining the research methodology and survey are described in previous publications [23–25].

Institutional review board (IRB) ethical approval was obtained from the University of Michigan, Boston University, and the ERES Converge IRB in Zambia.

## Sample

For the before-and-after cross-sectional survey evaluating the impact of the Core MWH Model (clinical trial #NCT02620436) multi-stage random sampling procedures were used with probability proportionate to population size [24]. The sample consisted of mothers who met the following inclusion criteria: (1) had given birth in the last 13 months (to obtain recent birth data and reduce recall bias), (2) 15 years of age or older, and (3) lived in a village that was 9.5 km or farther from one of the health care facilities included in our sample [24]. Details of the primary impact study including sampling frame, selection, assignment of study clusters, and protocol are reported elsewhere [23–26].

## Data collection

Locally trained research assistants recruited, consented, and enrolled participants from eligible households in the study [23]. Participants provided written informed consent, which was documented in writing or with a fingerprint and witness signature prior to beginning the survey [23]. For participants under the age of 18 years, child assent and guardian or husband (if over the age of 18 years) was obtained [23]. Research assistants were literate in the appropriate local languages and English. All had previous experience collecting quantitative data for research studies. Research assistants were trained in human subjects' protection and qualitative and quantitative data collection methods during a 5-day training [23–25]. Each household survey took approximately 45 minutes. Data were captured electronically on encrypted tablets using SurveyCTO Collect Software [23]. An in-depth description of measures used to assess change in use of the Core MWH Model, ANC and PNC attendance, family planning use, and vaccination rates is reported elsewhere [25]. In acknowledgment of their time, participants received a piece of local fabric as a token of appreciation [23]. Table 1 shows the measures used to construct the variables in our analysis.

**Table 1. Measures.**

| Dependent variables | Description |
|---|---|
| Frequency of antenatal care (ANC) | Number of times participant attended ANC at a health facility |
| Postnatal care visits (PNC) | Any postnatal checks after the first 24 hours following last birth |
| | Postnatal check approximately 3 days after last birth |
| | Postnatal check between 7 and 14 days after last birth |
| | Postnatal check approximately 6 weeks after last birth |
| Contraception/avoiding pregnancy | Currently using something or using any method to delay or avoid getting pregnant |
| Vaccinations for child | Child received any vaccinations |
| | Child received specific vaccinations at birth (i.e., BCG and OPV-0), 6 weeks (i.e., OPV-1, DTP-HepB-Hib-1, PCV, and Rotavirus), 10 weeks (i.e., OPV-2, DTP-HepB-Hib-2, PCV, and Rotavirus), and 14 weeks (i.e., OPV-3, DTP-HepB-Hib-3, and PCV). |
| **Independent variables** | |
| Use of the Core MWH Model | Use of the Core MWH Model for most recent birth |
| Baseline versus endline cohort | Participation in the survey prior to implementation of the Core MWH Model in 2016 (baseline) and after in 2018 (endline) |
| **Control variables** | Household size, marital status, number of births, age, and maternal educational level |

## Data analysis

The analysis is divided into two major sections. First, descriptive statistics are provided to assess differences (using Rao-Scott chi-square tests) between baseline and endline with respect to the dependent variables (ANC and PNC, family planning use, and vaccination rates), independent variable (i.e., use of the Core MWH Model), and control variables (i.e., household size, marital status, number of births, age, and educational level). Second, multivariable logistic regression were used to assess the association between use of the Core MWH Model and ANC and PNC. Additionally, these models also focus on differences between the baseline and endline cohorts with respect to the ANC and PNC. Accordingly, unadjusted odds ratios (OR), adjusted odds ratios (AOR) and 95% confidence intervals (95% CI) were provided to show these associations (i.e., differences between groups). Finally, all analyses use design-adjusted analytic techniques to account for clustering within each of the seven districts where the sample of participants were obtained. All analyses use Stata 15.0.

## Results

A total of 4711 mothers who recently gave birth to a child comprised the final overall sample (baseline n=2381; 50.5% and endline n=2330; 49.5%) with an average response rate of 88.8%. Of the mothers who were eligible but did not respond, 432 (8.4%) were unavailable, 95 (1.8%) refused participation, and 31 (0.6%) withdrew after beginning the survey or had incomplete surveys and were dropped from the analysis.

Table 2 provides the sociodemographic characteristics for the sample of mothers at both baseline and endline. Accordingly, approximately 40% of the mothers used the Core MWH Model for their most recent birth. Moreover, we see that the use of the Core MWH Model for mothers increased from 31.5% at baseline to 48.6% at endline (p<.01). Table 2 also provides the other sociodemographic characteristics across the two samples. It should be noted that both household size and mother's educational level were found to vary between baseline and endline (i.e., endline respondents had slightly more education and smaller household size).

The descriptive statistics and the crude OR for the ANC and PNC outcomes between baseline and endline are provided in Table 3. Overall, 64.9% (n=3052) of mothers attended four or more ANC visits, 52.1% (n=2455) attended any PNC visit, 7.7% (n=360) attended all PNC visits, 41.6% (n=1939) indicated actively avoiding pregnancy, 93.7% (n=4279) indicated that their child had received at least one vaccination, and 47.0% (n=1560) indicated that their child received all of their required vaccinations. Additionally, each of the antenatal/postnatal outcomes that mothers reported were higher at endline compared to baseline. For instance, 37.5% (n=602) of mothers indicated that their child received all of their vaccinations at baseline, while 56.0% (n=958) of mothers indicated that their child received all of their vaccinations at endline (p<.01).

Table 4 provides the crude odds ratio for the ANC and PNC outcomes of each factor. Of particular importance, mothers who stayed in a MWH during their most recent pregnancy had higher odds of attending four or more ANC visits (cOR = 1.65, 95% CI = 1.53. 1.79), attending any PNC visit (cOR = 1.44, 95% CI = 1.11. 1.87), attending all PNC visits (OR = 2.13, 95% CI = 1.61. 2.82), taking active measures to avoid pregnancy (cOR = 1.49, 95% CI = 1.21. 1.83), indicated that their child received at least one vaccination (cOR = 1.65, 95% CI = 1.29. 2.12), and indicating that their child received all of their vaccinations (cOR = 1.32, 95% CI = 1.07. 1.61) when compared to mothers who did not use the Core MWH Model during their most recent pregnancy. Additionally, mothers at endline also had higher odds of indicating each of these ANC and PNC outcomes when compared to the mothers at baseline.

The results of the multivariable logistic regression models are provided in Table 5. After controlling for survey period (i.e., baseline versus endline) and other sociodemographic

**Table 2. Sociodemographic characteristics between baseline and endline.**

|  | Total | Baseline | Endline | χ2 |
|---|---|---|---|---|
|  | (n = 4711) | (n = 2381) | (n = 2330) |  |
| **Used the Core MWH Model** |  |  |  | 20.72** |
| No | 2815(60.0%) | 1622 (68.5%) | 1193 (51.4%) |  |
| Yes | 1877(40.0%) | 747 (31.5%) | 1130 (48.6%) |  |
| **Household Size** |  |  |  | 5.05* |
| 1 to 3 people | 571(12.2%) | 275 (11.5%) | 296 (12.9%) |  |
| 4 to 6 people | 1953(41.7%) | 940 (39.5%) | 1013 (44.1%) |  |
| 7 or more people | 2156(46.1%) | 1166 (49.0%) | 990 (43.1%) |  |
| **Marital Status** |  |  |  | 4.71 |
| Married | 4097(87.6%) | 2092 (88.0%) | 2005 (87.1%) |  |
| Not Married | 582(12.4%) | 284 (12.0%) | 298 (12.9%) |  |
| **Number of Births** |  |  |  | 1.50 |
| At least 1 | 2246(48.3%) | 551 (23.2%) | 473 (21.7%) |  |
| 2 or 3 | 1920(41.3%) | 754 (31.8%) | 735 (33.7%) |  |
| 4 or more | 488(10.5%) | 1068 (45.0%) | 972 (44.6%) |  |
| **Age** |  |  |  | 0.22 |
| 15 to 19 | 821(17.6%) | 426 (18.0%) | 395 (17.2%) |  |
| 20 to 24 | 1525(32.7%) | 761 (32.1%) | 764 (33.3%) |  |
| 25 to 29 | 894(19.2%) | 454 (19.1%) | 440 (19.2%) |  |
| 30 to 34 | 752(16.1%) | 382 (16.1%) | 370 (16.1%) |  |
| 35 and older | 675(14.5%) | 349 (14.7%) | 326 (14.2%) |  |
| **Education** |  |  |  | 7.44* |
| No Education | 642(13.7%) | 362 (15.2%) | 280 (12.2%) |  |
| Some Primary | 1847(39.5%) | 968 (40.8%) | 879 (38.2%) |  |
| Completed Primary | 967(20.7%) | 476 (20.1%) | 491 (21.3%) |  |
| Some Secondary | 1118(23.9%) | 532 (22.4%) | 586 (25.5%) |  |
| Completed Secondary | 100(2.1%) | 36 (1.5%) | 64 (2.8%) |  |

*p < .05

**p < .01

***p < .001

χ2 = Rao-Scott chi-square test; Sample sizes may vary due to missing data.

characteristics, mothers who visited a MWH during their most recent pregnancy still had higher odds of attending four or more ANC visits (aOR = 1.48, 95% CI = 1.35. 1.62), attending all PNC visits (aOR = 2.02, 95% CI = 1.52. 2.68), taking active measures to avoid pregnancy (aOR = 1.33, 95% CI = 1.08. 1.63), indicated that their child received at least one vaccination (aOR = 1.30, 95% CI = 1.03. 1.65), and indicating that their child received all of their vaccinations (aOR = 1.20, 95% CI = 1.00. 1.44) when compared to mothers who did not use the Core MWH Model during their most recent pregnancy. Moreover, after controlling for Core MWH Model use and other sociodemographic characteristics, mothers who participated at endline also had higher odds of indicating each of these ANC and PNC outcomes (except for attending all PNC visits) when compared with mothers who participated at baseline.

## Discussion

This study examined the relationship between Core MWH Model use and accessing the recommended care for mothers and newborns in Zambia. Mothers who used a MWH had better

**Table 3. Antenatal and postnatal care outcomes between baseline and endline.**

|  | Total | Baseline | Endline | χ2 |
|---|---|---|---|---|
|  | (n = 4711) | (n = 2381) | (n = 2330) |  |
| **Attended four or more ANC visits** |  |  |  | 97.11*** |
| No | 1648 (35.1%) | 982 (41.4%) | 666 (28.6%) |  |
| Yes | 3052 (64.9%) | 1392 (58.6%) | 1660 (71.4%) |  |
| **Attended ANY PNC visit** |  |  |  | 41.46*** |
| No | 2256 (47.9%) | 1285 (54.0%) | 971 (41.7%) |  |
| Yes | 2455 (52.1%) | 1096 (46.0%) | 1359 (58.3%) |  |
| **Attended All PNC visit** |  |  |  | 11.20* |
| No | 4330 (92.3%) | 2226 (93.7%) | 2104 (90.9%) |  |
| Yes | 360 (7.7%) | 149 (6.3%) | 211 (9.1%) |  |
| **Avoiding Pregnancy** |  |  |  | 127.98*** |
| No | 2725 (58.4%) | 1580 (66.6%) | 1145 (50.0%) |  |
| Yes | 1939 (41.6%) | 792 (33.4%) | 1147 (50.0%) |  |
| **Child received ANY vaccinations** |  |  |  | 28.59** |
| No | 288 (6.3%) | 207 (8.9%) | 81 (3.6%) |  |
| Yes | 4279 (93.7%) | 2130 (91.1%) | 2149 (96.4%) |  |
| **Children received ALL vaccinations[1]** |  |  |  | 28.51** |
| No | 1757 (53.0%) | 1004 (62.5%) | 753 (44.0%) |  |
| Yes | 1560 (47.0%) | 602 (37.5%) | 958 (56.0%) |  |

*p < .05

**p < .01

***p < .001; χ2 = Rao-Scott chi-square test; Sample sizes may vary due to missing data.

[1]Analyses assessing whether a child received all required vaccinations only used the sample of mothers whose children where 14 weeks old or older. 72.2% of the mothers' most recent births included children 14 weeks or older (n = 3390).

ANC and PNC attendance, family planning use, and newborn vaccination rates at endline (even after controlling for various factors).

Regarding ANC, our findings are similar to a study testing the association between the presence of MWHs and personal and environmental factors that affect the use of MWHs [27]. In the cross-sectional study using an interviewer-administered questionnaire performed in the rural Kalomo district of Zambia, Sialubanje and colleagues found that respondents who used MWHs completed more ANC visits [27]. Although our cross-sectional study only assessed ANC in the reference pregnancy, and not subsequent pregnancies, it demonstrates a relationship between ANC and use of the Core MWH Model. This could be related to the community-based changes invoked by the Core MWH Model. The community mobilization process used to develop and then govern the MWHs have the potential to increase awareness and trust in the community about receiving clinical care (including ANC) with skilled providers. Furthermore, the physical MWH structure became a place of shelter for women traveling far distances for ANC care [28, 29].

In terms of PNC, our findings are consistent with those of researchers in Zambia's Southern and Eastern Provinces who used a time-series design to examine pre-post MWH intervention trends [30]. Researchers found the use of MWHs for PNC increased at one of two intervention sites even though both sites had dedicated space for PNC users [30]. Postnatal care is directly associated with place of birth and delivery by a skilled health provider [28, 31]. In line with this, use of MWHs has been associated with increased facility births and delivery by a skilled health provider [32, 33]. Hence, use of a MWH has the potential to facilitate use of PNC

**Table 4. Results of bivariate analysis assessing factors associated with antenatal and postnatal care outcomes.**

| | Attended four or more ANC visit | Attended ANY PNC visit | Attended All PNC visit | Avoiding Pregnancy | Child received ANY vaccinations | Children received ALL vaccinations[1] |
|---|---|---|---|---|---|---|
| | % OR 95% CI | % OR 95% CI | % OR 95% CI | % OR 95% CI | % OR 95% CI | % OR 95% CI |
| **Used the Core MWH Model** | | | | | | |
| No | 60.5% Reference | 48.5% Reference | 5.4% Reference | 37.7% Reference | 92.6% Reference | 44.1% Reference |
| Yes | 71.7% 1.65***(1.53-1.79) | 57.7% 1.44* (1.11-1.87) | 10.9% 2.13*** (1.61-2.82) | 47.5% 1.49** (1.21-1.83) | 95.4% 1.65** (1.29-2.12) | 51.1% 1.32* (1.08-1.61) |
| **Period** | | | | | | |
| Baseline | 58.6% Reference | 46.0% Reference | 6.3% Reference | 33.4% Reference | 91.1% Reference | 37.5% Reference |
| Endline | 71.4% 1.75***(1.52-2.02) | 58.3% 1.64*** (1.35-1.98) | 9.1% 1.49* (1.11-2.01) | 50.0% 1.99*** (1.71-2.32) | 96.4% 2.57*** (1.64-4.03) | 56.0% 2.12** (1.50-3.00) |
| **Household Size** | | | | | | |
| 1 to 3 people | 69.5% Reference | 57.1% Reference | 9.8% Reference | 45.1% Reference | 94.6% Reference | 50.0% Reference |
| 4 to 6 people | 65.4% 0.82 (0.67-1.01) | 53.3% 0.85* (0.75-0.97) | 8.3% 0.83 (0.63-1.09) | 46.4% 1.05 (0.75-1.48) | 94.2% 0.93 (0.44-1.91) | 46.3% 0.86 (0.65-1.14) |
| 7 or more people | 63.0% 0.75 (0.52-1.07) | 49.5% 0.73* (0.58-0.93) | 6.6% 0.65** (0.51-0.83) | 36.3% 0.69 (0.44-1.08) | 93.0% 0.76 (0.38-1.50) | 46.9% 0.88 (0.68-1.13) |
| **Marital Status** | | | | | | |
| Married | 65.7% Reference | 52.4% Reference | 7.7% Reference | 44.2% Reference | 93.9% Reference | 46.6% Reference |
| Not Married | 58.5% 0.73 (0.55-0.97) | 50.0% 0.90 (0.64-1.28) | 7.4% 0.96 (0.72-1.25) | 23.1% 0.38*** (0.30-0.48) | 92.1% 0.75 (0.44-1.26) | 50.5% 1.16 (0.86-1.58) |
| **Number of Births** | | | | | | |
| At least 1 | 64.8% Reference | 53.5% Reference | 7.9% Reference | 36.0% Reference | 92.6% Reference | 49.0% Reference |
| 2 or 3 | 63.7% 0.95 (0.81-1.10) | 51.8% 0.93 (0.75-1.15) | 7.2% 0.91 (0.72-1.15) | 46.7% 1.56*** (1.36-1.78) | 94.4% 1.33** (1.11-1.60) | 47.5% 0.94 (0.74-1.18) |
| 4 or more | 65.2% 1.01 (0.81-1.26) | 51.6% 0.92 (0.69-1.23) | 7.8% 0.99 (0.77-1.27) | 40.3% 1.20 (0.99-1.45) | 93.8% 1.21 (0.78-1.88) | 45.6% 0.87* (0.78-0.97) |
| **Age** | | | | | | |
| 15 to 19 | 64.1% Reference | 51.8% Reference | 6.6% Reference | 36.9% Reference | 92.3% Reference | 44.1% Reference |
| 20 to 24 | 64.8% 1.03 (0.84-1.27) | 52.1% 1.01 (0.74-1.39) | 8.7% 1.35 (0.88-2.08) | 45.8% 1.44*** (1.31-1.58) | 93.6% 1.21 (0.90-1.64) | 48.0% 1.16* (1.03-1.31) |
| 25 to 29 | 64.6% 1.02 (0.92-1.13) | 52.4% 1.02 (0.82-1.27) | 7.2% 1.09 (0.81-1.48) | 42.1% 1.24*** (1.13-1.36) | 94.3% 1.36 (0.81-2.27) | 47.5% 1.14 (0.87-1.49) |
| 30 to 34 | 65.3% 1.05 (0.83-1.34) | 53.2% 1.05 (0.85-1.31) | 6.5% 0.98 (0.58-1.65) | 41.3% 1.20 (0.87-1.66) | 94.6% 1.44 (0.83-2.50) | 43.8% 0.95 (0.88-1.09) |
| 35 and older | 65.3% 1.05 (0.87-1.27) | 50.7% 0.95 (0.63-1.45) | 8.7% 1.35 (0.97-1.89) | 37.0% 1.00 (0.89-1.12) | 93.7% 1.23 (0.70-2.16) | 51.0% 1.31* (1.03-1.68) |
| **Education** | | | | | | |
| No Education | 60.8% Reference | 49.1% Reference | 7.5% Reference | 36.6% Reference | 90.8% Reference | 44.2% Reference |
| Some Primary | 63.3% 1.11** (1.05-1.17) | 54.0% 1.21** (1.06-1.38) | 8.0% 1.08 (0.75-1.56) | 42.8% 1.29*** (1.18-1.43) | 94.7% 1.79* (1.18-2.70) | 48.1% 1.17 (0.96-1.42) |
| Completed Primary | 66.7% 1.29***(1.16-1.43) | 50.8% 1.07 (0.83-1.37) | 8.4% 1.13 (0.71-1.80) | 42.1% 1.26* (1.01-1.56) | 94.0% 1.57 (0.69-3.56) | 45.3% 1.04 (0.83-1.30) |
| Some Secondary | 67.8% 1.36** (1.15-1.60) | 51.3% 1.09 (0.87-1.37) | 6.4% 0.85 (0.48-1.49) | 41.8% 1.24 (0.97-1.58) | 93.3% 1.40 (0.88-2.23) | 47.0% 1.12 (0.91-1.37) |
| Completed Secondary | 69.0% 1.43* (1.01-2.02) | 57.0% 1.37 (0.96-1.96) | 9.0% 1.23 (0.57-2.65) | 44.4% 1.38 (0.66-2.91) | 96.0% 2.39 (0.68-8.36) | 62.0% 2.05 (0.85-4.96) |

*p < .05

**p < .01

***p < .001; [1]Analyses assessing whether a child received all required vaccinations only used the sample of mothers whose children where 14 weeks old or older. 72.2% of the mothers' most recent births included children 14 weeks or older (n = 3390).

**Table 5. Results of multivariable logistic regression models.**

| | Attended four or more ANC visit | Attended ANY PNC visit | Attended All PNC visit | Avoiding Pregnancy | Child received ANY vaccinations | Children received ALL vaccinations[1] |
|---|---|---|---|---|---|---|
| | aOR 95% CI | aOR 95% CI | aOR 95% CI | aOR 95% CI | aOR 95% CI | aOR 95% CI |
| | n = 4,481 | n = 4,482 | n = 4,474 | n = 4,477 | n = 4,392 | n = 3,208[1] |
| **Used the Core MWH Model** | | | | | | |
| No | Reference | Reference | Reference | Reference | Reference | Reference |
| Yes | 1.48***(1.35-1.62) | 1.31 (0.98-1.75) | 2.02***(1.52-2.68) | 1.33* (1.08-1.63) | 1.30* (1.03-1.65) | 1.20* (1.00-1.44) |
| **Period** | | | | | | |
| Baseline | Reference | Reference | Reference | Reference | Reference | Reference |
| Endline | 1.63***(1.38-1.92) | 1.56** (1.22-1.98) | 1.31 (0.90-1.90) | 1.92*** (1.69-2.18) | 2.56** (1.40-4.65) | 2.09** (1.39-3.13) |
| **Household Size** | | | | | | |
| 1 to 3 people | Reference | Reference | Reference | Reference | Reference | Reference |
| 4 to 6 people | 0.90 (0.73-1.10) | 0.85 (0.67-1.07) | 0.84 (0.52-1.38) | 0.91 (0.66-1.24) | 0.64 (0.30-1.37) | 0.89 (0.65-1.21) |
| 7 or more people | 0.81 (0.59-1.11) | 0.72 (0.52-1.00) | 0.60** (0.43-0.84) | 0.73* (0.54-0.98) | 0.54* (0.33-0.89) | 0.92 (0.71-1.18) |
| **Marital Status** | | | | | | |
| Married | Reference | Reference | Reference | Reference | Reference | Reference |
| Not Married | 0.74* (0.59-0.93) | 0.98 (0.68-1.40) | 1.26 (0.77-2.06) | 0.41*** (0.30-.581) | 1.04 (0.57-1.89) | 1.18 (0.85-1.63) |
| **Number of Births** | | | | | | |
| At least 1 | Reference | Reference | Reference | Reference | Reference | Reference |
| 2 or 3 | 0.86* (0.76-0.97) | 0.85 (0.60-1.20) | 0.79 (0.54-1.18) | 1.36* (1.00-1.85) | 1.32* (1.03-1.70) | 0.78 (0.51-1.19) |
| 4 or more | 0.99 (0.72-1.36) | 0.86 (0.47-1.56) | 1.08 (0.61-1.93) | 1.28 (0.72-2.27) | 1.09 (0.70-1.70) | 0.59* (0.41-0.86) |
| **Age** | | | | | | |
| 15 to 19 | Reference | Reference | Reference | Reference | Reference | Reference |
| 20 to 24 | 1.07 (0.94-1.23) | 1.09 (0.71-1.66) | 1.67 (0.99-2.82) | 1.01 (0.85-1.20) | 1.05 (0.80-1.38) | 1.45** (1.18-1.77) |
| 25 to 29 | 1.07 (0.86-1.33) | 1.18 (0.82-1.69) | 1.28 (0.83-1.98) | 0.85 (0.68-1.07) | 1.38 (0.80-2.39) | 1.73** (1.23-2.42) |
| 30 to 34 | 1.11 (0.79-1.54) | 1.28 (0.82-1.67) | 1.21 (0.61-2.40) | 0.94 (0.69-1.29) | 1.60 (0.94-2.73) | 1.65** (1.28-2.12) |
| 35 and older | 1.16 (0.90-1.49) | 1.18 (0.83-1.67) | 1.62 (0.95-2.73) | 0.78 (0.51-1.20) | 1.41 (0.66-3.00) | 2.20** (1.43-3.39) |
| **Education** | | | | | | |
| No Education | Reference | Reference | Reference | Reference | Reference | Reference |
| Some Primary | 1.06 (.974-1.16) | 1.16* (0.99-1.34) | 0.99 (0.62-1.57) | 1.22* (1.05-1.40) | 1.68* (1.13-2.49) | 1.13 (0.90-1.42) |
| Completed Primary | 1.22* (1.04-1.44) | 1.00 (0.81-1.24) | 1.05 (0.63-1.74) | 1.17 (0.98-1.41) | 1.57 (0.77-3.20) | 0.96 (0.69-1.33) |
| Some Secondary | 1.35***(1.19-1.52) | 1.03 (0.80-1.34) | 0.79 (0.47-1.34) | 1.19 (0.87-1.63) | 1.35 (0.87-2.10) | 0.99 (0.72-1.35) |
| Completed Secondary | 1.46* (1.09-1.95) | 1.28 (0.89-1.83) | 1.22 (0.58-2.55) | 1.44 (0.62-3.36) | 1.99 (0.56-7.05) | 1.59 (0.58-4.35) |

*p < .05

**p < .01

***p < .001; Sample sizes may vary due to missing data.

[1]Analyses assessing whether a child received all required vaccinations only used the sample of mothers whose children where 14 weeks old or older. 72.2% of the mothers' most recent births included children 14 weeks or older (n = 3390).

services after exposure to skilled attendants who counsel and encourage the mothers to come for PNC. The MWH also proves beneficial by providing shelter for mothers coming from distant places to wait for their PNC appointment near the health facility.

While no specific MWH articles have addressed family planning, we can situate our findings within the national Zambian context where 46% of women in rural areas report using

contraception.[12] In our study, fewer total mothers reported using contraception than the rural national average with 41.6% indicating actively avoiding pregnancy. However, at endline 50.0% (n = 1147) of mothers reported avoiding pregnancy up from 33.4% (n = 792) at baseline. This increase to half of mothers reporting using contraception highlights the potential positive impact of family planning health education in study communities. Other possible explanations for the increase in family planning use include the community mobilization and buy-in with the Core MWH Model and potentially an immediate uptake of postpartum family planning after use of the Core MWH Model.

To our knowledge, no other studies exploring MWHs have reported on childhood vaccination rates as an outcome. Vaccination coverage in our study was higher at endline than the Zambian national average. According to 2018 DHS data, in Zambia, 46.0% of children received all age-appropriate vaccinations compared to 56.0% in our study [13] In Zambia, children are considered to have received all basic vaccinations when they have received BCG vaccination, three doses of DPT vaccine (given as pentavalent), three doses of polio vaccine (excluding the polio vaccine given at birth), and a vaccination against measles (given as measles and rubella) [13].

In our study mothers who used the Core MWH Model had higher odds of attending more ANC and PNC visits along with taking active measures to avoid pregnancy and indicated that their child received all of their vaccinations. These increased contacts with the healthcare system have the potential to improve both maternal and newborn outcomes. This suggests a potential spillover effect regarding the MWH intervention – even if mothers did not go to a MWH, they could still be getting beneficial health information from mothers who did through knowledge sharing at the community/household level. Additionally, education provided to women staying at a MWH and the relationships they developed with skilled clinicians, could have improved their rates of return for PNC.

There are several components of the harmonized Core MWH Model that may have contributed to the investment, use, and spillover effects of the intervention. A component to support the Core MWH Model includes a financial sustainability strategy with multiple revenue streams [34]. This strategy involved creation of a financial sustainability model to fund the operations and maintenance of the MWH, with revenue derived from various sources, including community donations, health facility donations, and the creation of income generating activities. These income generating activities are managed by the MWH governance committees and function as social enterprises, generating revenue to operate and maintain the MWH.

Community mobilization is another component of the Core MWH Model potentially contributing to the uptake and spillover effects of the MWH intervention. The MWHs are community owned and operated and were developed with community mobilization and buy-in throughout the process [35]. The traditional leadership (chiefs and headmen) actively promote the use of MWHs at their community meetings [24]. Health facility staff promote the MWH at all ANC visits and maternal-newborn community health workers promote the use of MWHs during their routine outreach activities [24].

Another strength of this study design is that it meets the Cochrane review recommendation that well controlled trials are needed to continue to build evidence on MWH outcomes [22]. This study includes a population based household survey using a random design to gather and document a change at population level related to these variables.

## Limitations

As a generalizability limitation, the study was conducted in districts that received interventions prior to the start of the research as part of the SMGL initiative. The SMGL initiative

strengthened maternal health services with supply and demand side interventions to increase the timely use of quality maternity care therefore potentially increasing uptake of ANC and PNC attendance, family planning use, and childhood vaccinations [36]. Another limitation is that questions relied on participant's recall over the past 13 months, potentially affecting the accuracy of data collection. The results indicate that using a MWH at baseline or endline, regardless of the Core MWH Model, conferred significant benefits in regards to ANC attendance, PNC attendance, family planning use, and vaccination coverage. One possible explanation for this could be that MWHs within the intervention opened at various times over a one-year time frame. This staggered approach may not have allowed the endline data to capture the full effects of those MWHs that opened later in the study period. Also, the study focused only on individual-level factors and did not consider facility or system-level factors. Furthermore, wealth was not studied as a determinant and there is some evidence that MWHs are used preferentially by poorer women [37]. Finally, obstetric risk was not considered in the cross-sectional survey and in many contexts, women with higher obstetric risk are advised to access a MWH before birth.

## Conclusions

This study examined the relationship between Core MWH Model use and accessing the recommended care for mothers and newborns in Zambia. Our findings suggest an association between MWHs and improved ANC and PNC attendance, family planning use, and newborn vaccination outcomes. Future studies should evaluate whether availability of high quality services at facilities associated with MWHs is key to improving maternal and newborn outcomes. The three pillars of our Core MWH Model implemented in Zambia ensured high quality infrastructure, a formalized management structure, and access to skilled midwives. Maternity waiting homes have the potential to provide an enabling environment and access to high quality midwifery care for rural mothers in Zambia. Maternity waiting homes are a catalyst to improve visits with skilled nurses and midwives at health facilities to reach the most vulnerable mothers.

Additional research is needed past the endline time period of this study to fully understand the long-term impact of increased contacts with the healthcare system by mothers. Future research should also explore the sustainability of MWHs by rural Zambian communities after support from international partnerships is reduced.

## Author Contributions

**Conceptualization:** Michelle L. Munro-Kramer, Godfrey Biemba, Nancy Scott, Jody R. Lori.

**Data curation:** Philip T. Veliz, Xingyu Zhang.

**Formal analysis:** Julie M. Buser, Philip T. Veliz, Xingyu Zhang.

**Funding acquisition:** Nancy Scott, Jody R. Lori.

**Methodology:** Julie M. Buser, Michelle L. Munro-Kramer, Philip T. Veliz, Xingyu Zhang, Nancy Scott, Jody R. Lori.

**Project administration:** Nancy Lockhart, Godfrey Biemba, Thandiwe Ngoma, Nancy Scott, Jody R. Lori.

**Software:** Philip T. Veliz, Xingyu Zhang.

**Supervision:** Nancy Scott, Jody R. Lori.

**Validation:** Philip T. Veliz.

**Visualization:** Philip T. Veliz, Xingyu Zhang.

**Writing – original draft:** Julie M. Buser.

**Writing – review & editing:** Michelle L. Munro-Kramer, Philip T. Veliz, Xingyu Zhang, Nancy Lockhart, Godfrey Biemba, Thandiwe Ngoma, Nancy Scott, Jody R. Lori.

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
