## [Decision Letter · Decision Letter 0]

2 Dec 2020

PONE-D-20-31460

How maternity waiting home use influences attendance of antenatal and postnatal care

PLOS ONE

Dear Dr. Buser,

Thank you for submitting your manuscript to PLOS ONE. After careful consideration, we feel that it has merit but does not fully meet PLOS ONE’s publication criteria as it currently stands. Therefore, we invite you to submit a revised version of the manuscript that addresses the points raised during the review process.

We look forward to receiving your revised manuscript.

Kind regards,

Vijayaprasad Gopichandran

Academic Editor

PLOS ONE

Journal Requirements:

Reviewers' comments:

Reviewer's Responses to Questions

**Comments to the Author**

1. Is the manuscript technically sound, and do the data support the conclusions?

Reviewer #1: Partly

Reviewer #2: Partly

2. Has the statistical analysis been performed appropriately and rigorously? 

Reviewer #1: Yes

Reviewer #2: Yes

3. Have the authors made all data underlying the findings in their manuscript fully available?

Reviewer #1: Yes

Reviewer #2: Yes

4. Is the manuscript presented in an intelligible fashion and written in standard English?

Reviewer #1: Yes

Reviewer #2: Yes

5. Review Comments to the Author

Reviewer #1: The manuscript is properly written, and consider all scientific plausibility. General comment:the author need to mention about the structure, service/intervention provided to the mother who have used MWHs. what service the mother received through their stay in MWHs?. the study focused only individual factors, but not consider facility factors, and need to mention as a limitation, because for service outcome, facility factors are more relevant than individual factor. During the interpretation of the study result, MWHs users more likely higher in ANC, PNC and FP use, why? need explanation in the discussion part of the study. Avoiding pregnancy, there is a lot of intervention introduced to improve FP use, how this study sure MWHs contribute FP, perhaps, MWHs significantly contribute for immediate use of postpartum FP use. If the author consider this and other editorial issues, might be accepted for publication.

Reviewer #2: Thank you for the manuscript.

It describes a study in rural Zambia before and after a “Core MWH model” intervention. It reports on whether five maternal and newborn health system uptake indicators (≥4 ANC, any PNC, taking active measures to avoid pregnancy, child having received at least one vaccination, having received all vaccinations) are linked to MWH use, after controlling for sociodemographic characteristics and survey period. In adjusted analysis, the authors report that among mothers who had used a MWH before giving birth odds of the outcomes were increased.

The most important comment regards the conclusions of the study. The authors state that “mothers who went to a MWH had better ANC and PNC attendance, FP use … at endline” (line 244) and the title states that MWH use influences ANC and PNC. The outcomes and the exposure are linked, but the latter does is not necessarily responsible for the outcome. PNC, active measures to avoid pregnancy, and immunization are temporally after birth, so it appears plausible that exposure to midwives during a MWH stay may improve these outcomes. However, MWH use takes place before birth and ANC temporally precedes a MWH stay. Though ANC and MWH use were statistically linked, it is not clear how the conclusion that MWH influences ANC can be drawn from the results.

The second is about the intervention. The authors state that “Purpose of the study is the change in health system indicators before/after core MWH model (line 149 – 151)”. This model includes different components. Authors should refer to the “core MWH model” as the intervention, not just MWH use.

Additional comments

Throughout the manuscript, the authors should use give birth in place of “deliver”

Title – can the title be reworded to “are linked”? The authors could consider referring to the “core MWH model”

Abstract – this does not read easily at the moment. Methods in this section should be made clearer. Authors should include that before/after community surveys were carried out, as at present it is not mentioned.

Ln 84 – Authors state that “MWH may serve … especially those living furthest”. As distance from place of residence was not studied, this conclusion does not seem supported by the evidence in this study.

Introduction – can be more concise. Paragraph 3 is a bit difficult to follow. Further detail about the Zambian health system, where the MWH were located (in proximity to hospitals only or to other facilities too?) and which women received advise to stay in a MWH would enable readers to make comparisons to other contexts.

Line 124 – should read obstetric instead of pregnancy

Methods – authors sometimes refer to 2 outcomes (eg line 184) (ANC and PNC), in other parts of methods and results sections refer to five outcomes (Tables 1, 4 and 5). It should be clarified.

Results – this section is very clear.

Ln 217 – “…mothers who “visited” a MWH”. Do the authors refer to a MWH stay? The term visited makes it confusing to readers

Discussion

A discussion of how the authors interpret the finding of the link between ANC and MWH use should be included.

Women who are more concerned about their pregnancy, for example for a previous perinatal death, or for a multiple pregnancy, will access ANC early and use a MWH before birth. The two may be therefore be linked but one does not cause the other. More cautious language may be more suitable.

Do the authors refer to ANC in successive pregnancies? The information collected during the post-intervention survey (women who had given birth ≤13 months at the time of survey) does not seem to support this.

Also, community mobilization in the course of the intervention may have contributed to both ANC and MWH.

In addition, the link between MWH and facility birth is not discussed. Women who stay in a MWH will give birth in a facility – exposure to a skilled midwife will take place in both, and it is difficult in my view to separate the two effects.

In this respect, I believe the expression “they may serve as a catalyst” is great.

Wealth was not studied as a determinant. The authors should include this as a limitation, as there is some evidence that MWH are used preferentially by poorer women – see for example https://doi.org/10.1093/heapol/czx100

(I acknowledge that I was among authors of this publication)

Obstetric risk factors were not part of the control variables. In many contexts, women with higher obstetric risk are advised to access MWH before birth. This should be noted in greater detail among limitations.

6. PLOS authors have the option to publish the peer review history of their article (what does this mean?). If published, this will include your full peer review and any attached files.

Reviewer #1: No

Reviewer #2: No

---

## [Author Response · Author response to Decision Letter 0]

8 Jan 2021

January 7, 2021

Vijayaprasad Gopichandran

Academic Editor

PLOS ONE

Title: "How maternity waiting home use influences attendance of antenatal and postnatal care” (PONE-D-20-31460)

Dear Dr. Gopichandran and Editorial Team:

We are very pleased to have the opportunity to revise our original manuscript for publication in PLOS ONE. We addressed each of the journal requirements along with reviewer’s suggestions and made the necessary revisions and modifications. Responses are provided below.

Journal Requirements:

Comment #1: Please ensure that your manuscript meets PLOS ONE's style requirements, including those for file naming. The PLOS ONE style templates can be found at

Response #1: We updated the manuscript to ensure that it meets PLOS ONE's style requirements, including those for file naming.

Comment #2: Please provide additional details regarding participant consent. In the ethics statement in the Methods and online submission information, please ensure that you have specified what type you obtained (for instance, written or verbal, and if verbal, how it was documented and witnessed). If your study included minors, state whether you obtained consent from parents or guardians. If the need for consent was waived by the ethics committee, please include this information.

 Response #2: We provided the additional details regarding participant consent as requested. 

Comment #3: Your ethics statement should only appear in the Methods section of your manuscript. If your ethics statement is written in any section besides the Methods, please delete it from any other section.

Response #3: We deleted the ethics statement from the title page. 

 

Reviewer(s)' comments:

Reviewer #1

Comment #1: The manuscript is properly written, and consider all scientific plausibility. General comment:the author need to mention about the structure, service/intervention provided to the mother who have used MWHs. what service the mother received through their stay in MWHs?. the study focused only individual factors, but not consider facility factors, and need to mention as a limitation, because for service outcome, facility factors are more relevant than individual factor. 

Response #1: We sincerely thank the reviewer for this comment. We added language to mention facility factors about the structure and service/intervention provided to the mothers who used MWHs. We also mentioned in the Limitations section that the study focused only individual factors, but did not consider facility factors. 

Comment #2: During the interpretation of the study result, MWHs users more likely higher in ANC, PNC and FP use, why? need explanation in the discussion part of the study. 

Response #2: Thank you for your comment. We have expanded our discussion to include a more robust analysis of the relationship between the Core MWH Model and ANC, PNC, and family planning use including potential explanataions.

Comment #3: Avoiding pregnancy, there is a lot of intervention introduced to improve FP use, how this study sure MWHs contribute FP, perhaps, MWHs significantly contribute for immediate use of postpartum FP use. 

Response #3: We appreciate this suggestion for improvement and revised the text to include that perhaps MWHs significantly contribute for immediate use of postpartum FP use.

Reviewer #2

Comment #1: The authors state that “mothers who went to a MWH had better ANC and PNC attendance, FP use … at endline” (line 244) and the title states that MWH use influences ANC and PNC. The outcomes and the exposure are linked, but the latter does is not necessarily responsible for the outcome. PNC, active measures to avoid pregnancy, and immunization are temporally after birth, so it appears plausible that exposure to midwives during a MWH stay may improve these outcomes. However, MWH use takes place before birth and ANC temporally precedes a MWH stay. Though ANC and MWH use were statistically linked, it is not clear how the conclusion that MWH influences ANC can be drawn from the results.

Response #1: Thank you for this comment. We have added some additional details throughout the manuscript to help illustrate this relationship focusing specifically on the fact that women did use the MWH as a shelter when seeking ANC care and that the Core MWH Model involved a community mobilization and community ownership process (lines 151-166) with the MWHs that may have influenced community members awareness and desires to seek ANC care with skilled healthcare providers (lines 279-285).

Comment #2: The authors state that “Purpose of the study is the change in health system indicators before/after core MWH model (line 149 – 151)”. This model includes different components. Authors should refer to the “core MWH model” as the intervention, not just MWH use.

Response #2: We thank the review for this comment and updated the text to refer to the “core MWH model” as the intervention, not just MWH use.

Comment #3: Throughout the manuscript, the authors should use give birth in place of “deliver”

Response #3: We revised language throughout the manuscript to use give birth in place of “deliver”. 

Comment #4: Title – can the title be reworded to “are linked”? The authors could consider referring to the “core MWH model”

Response #4: Thank you for this comment. We reworded the title as suggested. 

Comment #5: Abstract – this does not read easily at the moment. Methods in this section should be made clearer. Authors should include that before/after community surveys were carried out, as at present it is not mentioned.

Response #5: We inserted language in the abstract to clarify the Methods section and include that before/after community surveys were carried out. 

Comment #6: Ln 84 – Authors state that “MWH may serve … especially those living furthest”. As distance from place of residence was not studied, this conclusion does not seem supported by the evidence in this study.

Response #6: We appreciate your observation and deleted this conclusion from the manuscript. 

Comment #7: Introduction – can be more concise. Paragraph 3 is a bit difficult to follow. Further detail about the Zambian health system, where the MWH were located (in proximity to hospitals only or to other facilities too?) and which women received advise to stay in a MWH would enable readers to make comparisons to other contexts.

Response #7: We made the introduction more concise and reworded paragraph 3 to make it easier to follow. We inserted language about the Zambian health system (lines 102-105) as well as additional details about the MWH location and users (lines 148-166). 

Comment #8: Line 124 – should read obstetric instead of pregnancy

Response #8: We reworded the text as suggested.

Comment #9: Methods – authors sometimes refer to 2 outcomes (eg line 184) (ANC and PNC), in other parts of methods and results sections refer to five outcomes (Tables 1, 4 and 5). It should be clarified.

Response #9: Thank you, we clarified the text as suggested. 

Comment #10: Results – this section is very clear.

Response #10: We sincerely thank the reviewer for sharing this observation. 

Comment #11: Ln 217 – “…mothers who “visited” a MWH”. Do the authors refer to a MWH stay? The term visited makes it confusing to readers

Response #11: We clarified the language to refer to a “MWH stay”. 

Comment #12: Discussion – A discussion of how the authors interpret the finding of the link between ANC and MWH use should be included. Women who are more concerned about their pregnancy, for example for a previous perinatal death, or for a multiple pregnancy, will access ANC early and use a MWH before birth. The two may be therefore be linked but one does not cause the other. More cautious language may be more suitable.

Response #12: Thank you for this comment. We have tempered our language in the discussion section and made sure all limitations are included.

Comment #13: Do the authors refer to ANC in successive pregnancies? The information collected during the post-intervention survey (women who had given birth ≤13 months at the time of survey) does not seem to support this.

Response #13: The ANC visits were those that occurred in the reference visit (within the last 13 months), not any subsequent pregnancies. This has been clarified in the text (lines 279-281) 

Comment #14: Also, community mobilization in the course of the intervention may have contributed to both ANC and MWH.

Response #14: Thank you, we noted this in the Discussion section. 

Comment #15: In addition, the link between MWH and facility birth is not discussed. Women who stay in a MWH will give birth in a facility – exposure to a skilled midwife will take place in both, and it is difficult in my view to separate the two effects. In this respect, I believe the expression “they may serve as a catalyst” is great.

Response #15: Thank you for this comment and the agreement with our use of langauge. We have discussed the relationship between MWH, facility births, and skilled providers in lines 290-294.

Comment #16: Wealth was not studied as a determinant. The authors should include this as a limitation, as there is some evidence that MWH are used preferentially by poorer women – see for example https://doi.org/10.1093/heapol/czx100 (I acknowledge that I was among authors of this publication)

Response #16: We appreciate this suggestion for improvement and updated the limitations to include that wealth was not studied as a determinant and inserted the example citation.

Comment #17: Obstetric risk factors were not part of the control variables. In many contexts, women with higher obstetric risk are advised to access MWH before birth. This should be noted in greater detail among limitations.

Response #17: We thank the review for this comment and noted in greater detail among limitations that in many contexts, women with higher obstetric risk are advised to access MWH before birth.

Sincerely,

The Authors

---

## [Editor Report · Decision Letter 1]

11 Jan 2021

How maternity waiting home use influences attendance of antenatal and postnatal care

PONE-D-20-31460R1

Dear Dr. Buser,

We’re pleased to inform you that your manuscript has been judged scientifically suitable for publication and will be formally accepted for publication once it meets all outstanding technical requirements.

Kind regards,

Vijayaprasad Gopichandran

Academic Editor

PLOS ONE
---

## [Editor Report · Acceptance letter]

13 Jan 2021

PONE-D-20-31460R1 

How maternity waiting home use influences attendance of antenatal and postnatal care 

Dear Dr. Buser:

I'm pleased to inform you that your manuscript has been deemed suitable for publication in PLOS ONE. Congratulations! Your manuscript is now with our production department. 

Kind regards, 

on behalf of

Dr. Vijayaprasad Gopichandran 

Academic Editor

PLOS ONE